environmental chemistry

microaerobic, micro-electrolysis, waterfall aeration, microbial community

**Authors for correspondence:**
Qiang Lin
e-mail: linqianggroup@163.com
Yu-hong Feng
e-mail: fengyuhong@hainanu.edu.cn

This article has been edited by the Royal Society of Chemistry, including the commissioning, peer review process and editorial aspects up to the point of acceptance.

# Pollutant removal from municipal sewage by a microaerobic up-flow oxidation ditch coupled with micro-electrolysis

Zhen-dong Zhao[1,2], Qiang Lin[1], Yang Zhou[3], Yu-hong Feng[2], Qi-mei Huang[3] and Xiang-hui Wang[1]

[1]Key Laboratory of Water Pollution Treatment and Resource Reuse of Hainan Province, Key Laboratory of Natural Polymer Functional Material of Haikou City, College of Chemistry and Chemical Engineering, Hainan Normal University, Haikou 571158, People's Republic of China
[2]Analytical and Testing Center, and [3]School of Chemical Engineering and Technology, Hainan University, Haikou 570228, People's Republic of China

QL, 0000-0002-4971-3293

The development of efficient and low-cost wastewater treatment processes remains an important challenge. A microaerobic up-flow oxidation ditch (UOD) with micro-electrolysis by waterfall aeration was designed for treating real municipal wastewater. The effects of influential factors such as up-flow rate, waterfall height, reflux ratio, number of stages and iron dosing on pollutant removal were fully investigated, and the optimum conditions were obtained. The elimination efficiencies of chemical oxygen demand (COD), ammonia nitrogen ($NH_4^+$-N), total nitrogen (TN) and total phosphorus (TP) reached up to $84.33 \pm 2.48\%$, $99.91 \pm 0.09\%$, $93.63 \pm 0.60\%$ and $89.27 \pm 1.40\%$, respectively, while the effluent concentrations of COD, $NH_4^+$-N, TN and TP were $20.67 \pm 2.85$, $0.02 \pm 0.02$, $1.39 \pm 0.09$ and $0.27 \pm 0.02$ mg l$^{-1}$, respectively. Phosphorous removal was achieved by iron–carbon micro-electrolysis to form an insoluble ferric phosphate precipitate. The microbial community structure indicated that carbon and nitrogen were removed via multiple mechanisms, possibly including nitrification, partial nitrification, denitrification and anammox in the UOD.

## 1. Introduction

Eutrophication due to the excess nutrients emission including nitrogen and phosphorus into surface water sources has

triggered serious threats to human health, the environment and even ecosystems [1–3]. Therefore, removing nitrogen and phosphorous from wastewater is necessary for tackling widespread water environmental problems [4,5]. It is well known that nitrogen removal can be conducted via conventional nitrification and denitrification. These approaches have some negative effects including requirements of both large-scale structures and high costs for the aeration [6]. In addition, the extra carbon sources required for heterotrophic denitrification cause substantial excess sludge production [7]. To further address the shortcomings of those traditional nitrogen removal methods, new principles and strategies have been proposed, such as partial nitrification–denitrification [8–11], anaerobic ammonium oxidation (anammox) [12–14], and simultaneous nitrification and denitrification (SND) [15–17]. These approaches are based on the fact that ammonium-oxidizing bacteria (AOB) convert ammonium into an intermediary compound, nitrite, which is then transformed to nitrogen gas by autotrophic denitrification, heterotrophic denitrification and anammox [18,19]. These processes have been proven to be more energy-efficient than existing alternatives.

To accumulate nitrite, nitrite-oxidizing bacteria (NOB) must be suppressed with the controllable dissolved oxygen (DO) at specifically low concentrations [20,21]. The reasons for this prerequisite are that the oxygen affinity of NOB is lower than that of AOB and that the saturation coefficients of oxygen gas obtained in Monod kinetics for nitritation and nitration were 0.3 mg l$^{-1}$ and 1.1 mg l$^{-1}$, respectively [22]. Hence, the microaerobic treatment process is a proposed method for treating wastewater and is carried out under a low DO concentration (less than or equal to 1.0 mg l$^{-1}$) [23], resulting in energy efficiency and the minimization of excess sludge production. Moreover, compared with the conventional biological process of nitrogen removal, the microaerobic treatment process can eliminate carbon and nitrogen in a single reactor because of the coexistence of different microorganisms. These microorganisms occupy and proliferate at different depths of flocs and granules due to their varying oxygen affinities, causing an oxygen gradient distribution. Observations have revealed that AOB, NOB and heterotrophic aerobic bacteria occupy the surface of flocs or granules, while microaerophilic bacteria and anaerobes dominate the inner side [24]. Therefore, carbon and nitrogen can be rapidly reduced in a microspace with enhanced removal efficiency during the microaerobic processes.

To date, studies treating wastewater with mechanical aeration in an oxygen-limited sequencing batch reactor (SBR) have been published [20–22,25,26]. However, it seems that the efficiency of large-scale microaerobic treatment by using mechanical aeration is still unsatisfactory [27]. The oxygen transfer efficiency of mechanical aeration can be influenced by many factors, such as internal clogging and external fouling of diffusers, leading to poor stability [28,29]. Besides, there is also a problem of inaccurate DO concentrations, and current measurement approaches cannot reflect the real DO at each position rapidly and precisely [29]. Obviously, it is necessary to invent a new aeration method instead of mechanical methods to obtain stable and precise low-oxygen conditions.

During the water falling, oxygen can dissolve into the water via the air–water interface and then spread, causing reoxygenation of DO in the water [24]. The problems of mechanical aeration can be avoided using waterfall aeration, which has been proven in our built-up sewage treatment station. There are a few studies that treated low-chemical oxygen demand (COD)/total nitrogen (TN) ratio synthetic wastewater using a single microaerobic reactor with waterfall aeration [30,31]. However, up-flow oxidation ditches (UODs), which are multi-stage serial intermittent up-flow sludge reactors, have drawn less attention in the treatment of real municipal sewage. Each stage of a UOD contains a waterfall aeration chamber and an up-flow chamber. Reoxygenation is obtained through the process of water falling from the top of the waterfall aeration chamber. The concentration of DO can be adjusted by many factors, including the up-flow rate, waterfall height, reflux ratio and the number of stages, whose influences on pollutant removal from real domestic wastewater are not fully understood.

Moreover, the presence of phosphorus can be completely removed through the chemical precipitation. Usually, iron or aluminium salts are added to form insoluble precipitates that can be sedimented or filtered. For example, ferric ions can directly react with orthophosphate chemicals to yield the precipitates of ferric phosphates [15,32,33]. Taking the cost of ferric iron salts into account, the inexpensive by-product iron scurf can generate ferrous ions through iron–carbon micro-electrolysis and then the formed ferrous ions are oxidized into ferric ions under aeration conditions [34–36]. Hence, the removal of phosphorus in UODs can be promoted by iron–carbon micro-electrolysis.

In this work, a new type of microaerobic UOD coupled with iron–carbon micro-electrolysis was designed. To better understand the operation mechanism and obtain a greater simultaneous removal efficiency, the effects of the up-flow rate, waterfall height, reflux ratio, number of stages and iron dosing on carbon, nitrogen and phosphorous removal were fully investigated during the purification of municipal wastewater. Furthermore, the existence of a microbial community was examined during

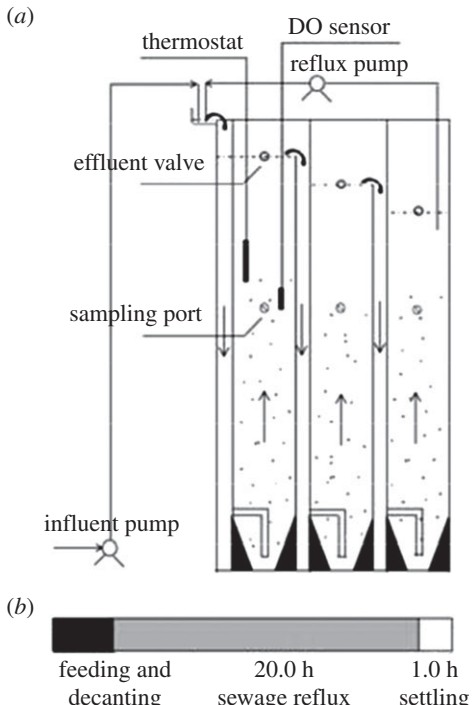

(a)

(b)

feeding and decanting | 20.0 h sewage reflux | 1.0 h settling

**Figure 1.** Schematic diagram of the UOD reactor.

the treatment. We expected these results could provide helpful insights and suggestions for the further design and industrial applications of UOD reactors for pollutant removal.

# 2. Material and methods

## 2.1. Reactor and experimental set-up

The specific structure of the UOD reactor used in this work is shown in figure 1, which was a multi-stage serial intermittent up-flow sludge reactor under microaerobic conditions. It consisted of a series of waterfall aeration chambers and up-flow chambers and their bottoms were connected by a pipeline. The open roof supplied oxygen for dissolution in the falling water. After a settling period of 1 h, raw sewage was pumped up to the first waterfall aeration chamber to achieve reoxygenation and then transferred into the up-flow chamber through the passage at the bottom. After rising to the top, the effluent overflowed into the second waterfall aeration chamber to repeat the process. Finally, an effluent line connected to the last up-flow chamber permitted the outflow of the treated sewage. After feeding (reflux ratio, 3 : 1), the wastewater was returned to the top of the first waterfall aeration chamber by a peristaltic pump. The reflux operation was continued for 20 h until the reaction was completed. The height of the reactor was 1.2 m. To estimate the effects of the up-flow rate, waterfall height, reflux ratio, number of stages and iron dosing on pollutant removal, this research used a series of UODs with the same design except for the height of falling water. Specifically, the UODs were costumed using the Plexiglas material (volume 24 l with a waterfall height of 10 cm or volume 22 l with a waterfall height of 14 cm). During the experiment, the DO concentration was recorded using a portable HACH DO monitor (HQ30D, HACH, Loveland, CO, USA).

## 2.2. Wastewater and seed sludge

These municipal wastewater and seed sludge samples used for various tests were obtained from the aerated grit chamber and the oxic tank of the Haikou Wastewater Treatment Plant (WWTP) (from Hainan Province, China), with a treatment ability of 300 000 $m^3 d^{-1}$ for the municipal wastewater. The main chemical parameters of these influents are listed in table 1.

**Table 1.** Characteristics of influent municipal sewage and seed sludge used in this research.

| name | parameter | mean (minimum and maximum) |
|---|---|---|
| sewage | COD (mg l$^{-1}$) | 95–158 |
| | BOD (mg l$^{-1}$) | 42–79 |
| | NH$_4^+$-N (mg l$^{-1}$) | 17.06–22.65 |
| | TN (mg l$^{-1}$) | 18.1–25.0 |
| | TP (mg l$^{-1}$) | 1.9–3.0 |
| | pH | 6.85–7.26 |
| the initial biomass in the UODs | MLSS (g l$^{-1}$) | 2.85–2.97 |
| | MLVSS (g l$^{-1}$) | 0.74–0.85 |

## 2.3. Analytical methods

Influent samples have a larger concentration of organic matrix than effluent samples, reducing ion column and suppressor life. The concentrations of COD, NH$_4^+$-N, TN and TP were detected directly by a multi-parameter portable colorimeter (DR900, HACH). An ion chromatography (IC) was employed to measure the concentrations of NO$_2^-$-N, NO$_3^-$-N, TP and NH$_4^+$-N in the effluent samples. Anions (NO$_2^-$, NO$_3^-$ and PO$_4^{3-}$) were determined on an ICS-1500 ion chromatograph (DIONEX, Sunnyvale, CA, USA), which was equipped with an IonPac AS9-HC (250 × 4 mm) column, an anion suppressor and a conductivity detector. The flow eluent was made of 9 mM NaCO$_3$ with a rate of 1 ml min$^{-1}$ and the temperatures of both column and detector were kept at 35°C. The suppressor current was maintained at 50 mA. NH$_4^+$ was analysed on a CIC 500 IC (Shenghan, Qingdao, Shangdong Province, China) with a universal cation column (100 × 4.6 mm) and a conductivity detector. Oxalate (2.5 mM) was employed as the mobile phase with a rate of 1.0 ml min$^{-1}$, in which the temperatures of the column and detector were 40 °C. To eliminate matrix effects on IC, all samples were diluted four-fold with DI water and then prepared using an SPE-C18 column. For anion analysis, samples were also passed through an SPE-H column. To optimize the performance, a total of 60 samples were collected for each experimental condition and a total of 378 samples were collected.

Under working condition, the specific concentration of suspended substance (SS) in the effluent is very low. Therefore, we presumed that the organic nitrogen content in the effluent is close to zero and can be ignored. In this case, the TN of effluent and its specific removal efficiency were calculated by equations (2.1) and (2.2), respectively

$$TN_{effluent} = NH_3-N + NO_2-N + NO_3-N \tag{2.1}$$

and

$$Removal\ efficiency = \frac{influent\ concentration - effluent\ concentration}{influent\ concentration \times 100\%} \tag{2.2}$$

## 2.4. Operation of the UODs

Seed sludge was first acclimated for 60 days. Before performing each different experiment, a 7-day adjustment period for each UOD was needed. During the period of acclimation and adjustment, the reflux ratio was determined as 2 m h$^{-1}$ and the up-flow rate was set to 3.5 m h$^{-1}$. To investigate the influences on the COD and nitrogen removal efficiencies, the DO concentration was slightly changed by altering the up-flow rate (2.5, 3.0, 3.5 and 4.0 m h$^{-1}$), waterfall height (10 and 14 cm), reflux ratio (3, 2, 1) and the number of stages (one, two and three). Besides, the effect on phosphorus removal was examined by adding iron scurf (1, 2 and 4 g l$^{-1}$) and changing the up-flow rate. The specific hydraulic retention time was set up to 20 h, in which the temperature was maintained at 30 ± 2°C.

## 2.5. DNA extraction, PCR amplification and Illumina sequencing

The seed (S0) and activated (S1–3, corresponding to the first, second and third stages of the UOD reactor) sludge were both sampled. DNA extractions were performed using a NucleoSpin 96 Soi DNA extraction

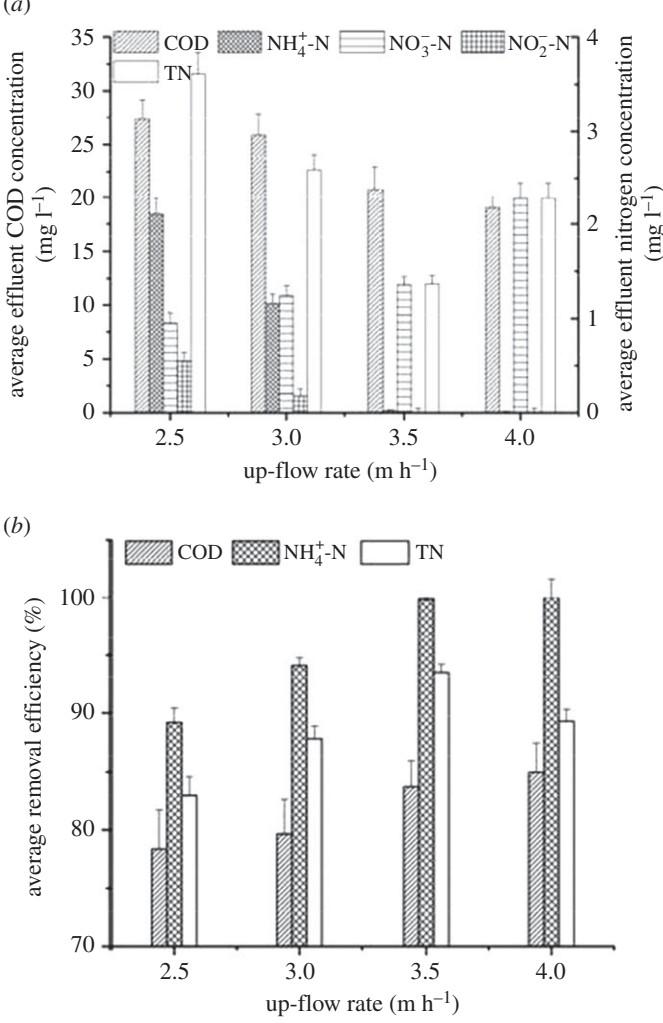

**Figure 2.** COD and nitrogen removal performances with different up-flow rates: (*a*) average effluent concentration and (*b*) average removal efficiency.

kit (purchased from Machery-Nagel GmbH & Co. KG, Düren, Germany). Bacterial 16S rRNA genes for the V3–V4 regions were amplified with the primer pair 338F (5′-ACTCCTACGGGAGGCAGCA-3′) and 806R (5′-GGACTACHVGGGTWTCTAAT-3′). The corresponding compositions of PCR were determined through the Illumina Hiseq2500 platform (Shanghai Biotree, China) for the paired-end sequencing.

# 3. Results and discussion

## 3.1. COD and nitrogen removal efficiency of the UOD system

### 3.1.1. Effects of up-flow rate on COD and nitrogen removal

Four UOD reactors (waterfall height: 10 cm) were applied to treat real municipal wastewater under different up-flow rates, and the reflux ratio was set to 2. Electronic supplementary material, figure S1 shows the COD and nitrogen removal performances over a 60-day operational period, and figure 2 demonstrates the averages within this period.

When the up-flow rate increased from 2.5 to 4.0 m h$^{-1}$, the average removal efficiencies for COD and NH$_4^+$-N were increased gradually. The average COD removal efficiency increased from 78.38 ± 3.37% to 84.98 ± 2.44%, and the average effluent COD concentration decreased from 27.40 ± 1.78 to 19.05 ± 1.05 mg l$^{-1}$. Moreover, the average removal efficiency for NH$_4^+$-N was increased from 89.22 ± 1.21% to 100.00%, and the average effluent NH$_4^+$-N concentration decreased from 2.11 ± 0.17 mg l$^{-1}$ to 0. The average effluent NO$_3^-$-N reached a maximum concentration of 2.28 ± 0.17 mg l$^{-1}$, resulting from the

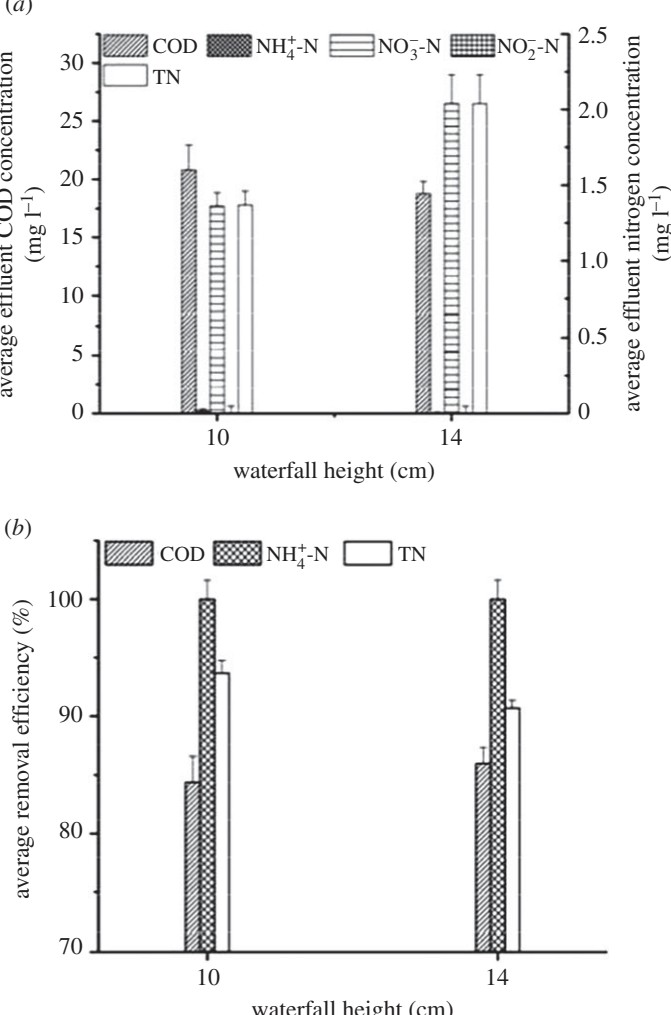

**Figure 3.** COD and nitrogen removal performances with different waterfall heights: (*a*) average effluent concentration and (*b*) average removal efficiency.

rising up-flow rate, while the concentration of $NO_2^-$-N was dropped from $0.55 \pm 0.09$ mg l$^{-1}$ to 0. Under different up-flow rates, the average TN removal efficiency first increased to a maximum of $93.47 \pm 0.72\%$ and then decreased as the up-flow rate increased further. Additionally, the average effluent TN was decreased from $3.61 \pm 0.23$ to $1.37 \pm 0.09$ mg l$^{-1}$ and then rose to $2.28 \pm 0.17$ mg l$^{-1}$.

Based on the trend observed in the average removal efficiencies for $NH_4^+$-N, UOD with falling water aeration could provide enough oxygen for the oxidation of ammonia. The DO concentration could be slightly adjusted by changing the up-flow rate. The activity of NOB was inhibited under lower DO concentrations [37], leading to an accumulation of nitrite, promoting the combination between partial denitrification and anammox. To achieve the maximum TN removal, the up-flow rate was maintained at 3.5 m h$^{-1}$ in the following experiments.

### 3.1.2. Effects of waterfall height on COD and nitrogen removal

Experiments were performed to study the influences of waterfall height on COD and nitrogen removal by using two UOD reactors (up-flow rate of 3.5 m h$^{-1}$ and reflux ratio of 2). Electronic supplementary material, figure S2 shows the results of continuous operation for 60 days. Figure 3 displays the 60-day averages in removal efficiency and effluent concentration.

At waterfall heights of 10 and 14 cm, the average elimination efficiency for COD was always higher than $84.38 \pm 2.21\%$, and the average effluent COD concentrations were $20.82 \pm 2.14$ mg l$^{-1}$ and $18.75 \pm 1.07$ mg l$^{-1}$, respectively. However, ammonium could not be detected. With an increase in waterfall height, the average effluent concentration of $NO_3^-$-N rose from $1.36 \pm 0.09$ to $2.04 \pm 0.19$ mg l$^{-1}$ and $NO_2^-$-N was almost constant at zero. Compared with the result at a waterfall height of 10 cm, the

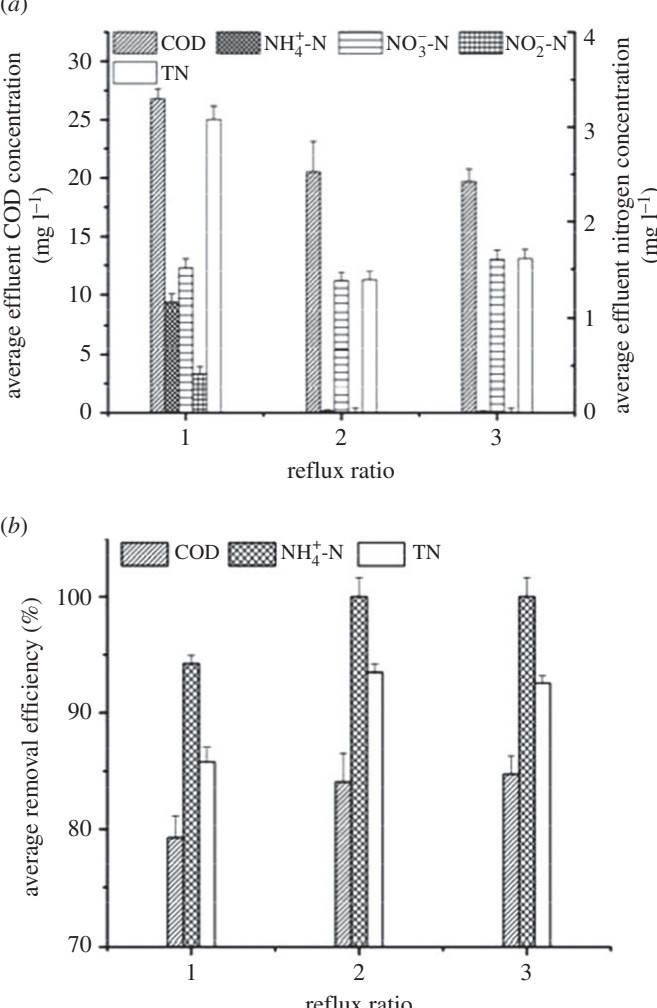

**Figure 4.** COD and nitrogen removal performances at different reflux ratios: (*a*) average effluent concentration and (*b*) average removal efficiency.

average TN removal efficiency at a height of 14 cm dropped to $90.70 \pm 1.09\%$, and the average effluent TN concentration jumped to $2.04 \pm 0.19$ mg l$^{-1}$. Our results proved that the aeration efficiency of falling water showed a positive correlation with the enhancement of waterfall height. At the falling height of 14 cm, a higher DO concentration ($0.25–0.51$ mg l$^{-1}$) was provided, causing the recovery of NOB activity and an increase in $NO_3^-$-N concentration. The aeration by the lower waterfall height of 10 cm was sufficient to achieve an appropriate DO concentration ($0.19–0.40$ mg l$^{-1}$) during the whole process.

### 3.1.3. Effects of reflux ratio on COD and nitrogen removal

The influences on COD and nitrogen removal were further investigated at different reflux ratios (3, 2 and 1) in three UOD reactors over a 60-day operational period (electronic supplementary material, figure S3) under a waterfall height of 10 cm and up-flow rate of 3.5 m h$^{-1}$. The average removal efficiency and effluent concentration values are shown in figure 4.

The average removal efficiency for $NH_4^+$-N was dropped to $94.23 \pm 0.72\%$ at a reflux ratio of 1, in contrast with 100% at reflux ratios of 2 and 3. A lower reflux ratio provided a lower DO concentration that resulted in incomplete oxidation of ammonium. The average COD removal efficiency slowly improved, ranging from $79.28 \pm 1.84$ to $84.75 \pm 1.57\%$, and the average COD concentration decreased from $26.77 \pm 0.89$ to $19.70 \pm 1.08$ mg l$^{-1}$ when the reflux ratio was varied from 1 to 3. Due to insufficient oxygen supply at a low reflux ratio of 1, the accumulation of $NO_2^-$-N was approximately $0.41 \pm 0.08$ mg l$^{-1}$. By contrast, $NO_2^-$-N was not detected in the effluent under the higher reflux ratios of 2 and 3. The average concentration of effluent $NO_3^-$-N in all three reactors was approximately

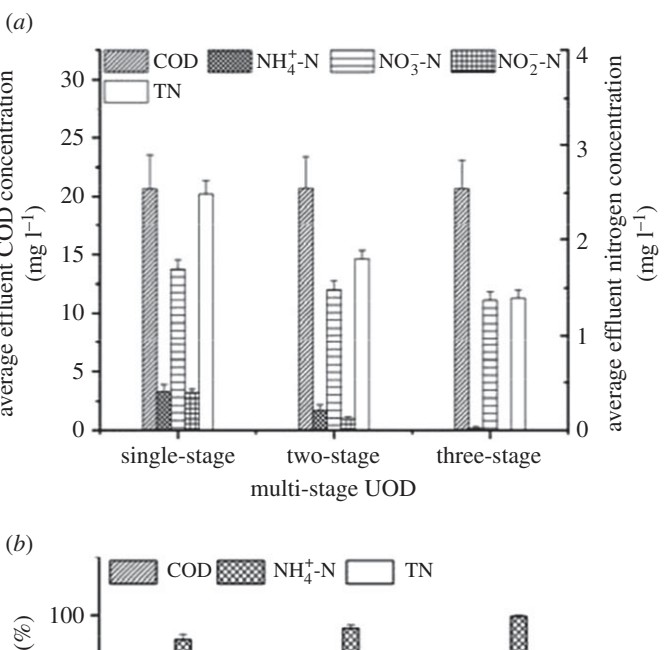

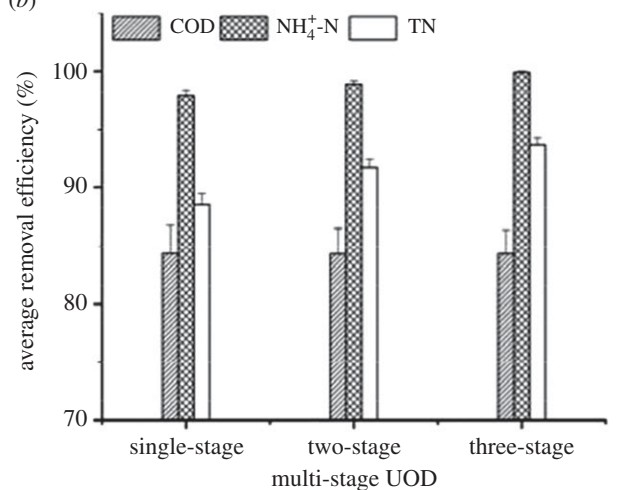

**Figure 5.** COD and nitrogen removal performances of single-, two- and three-stage UODs: (*a*) average effluent concentration and (*b*) average removal efficiency.

$1.51 \pm 0.10$ mg l$^{-1}$. As the reflux ratio increases, the average TN removal efficiency was initially enhanced from $85.80 \pm 1.19$ to $93.48 \pm 0.70\%$ and then decreased to $92.55 \pm 0.67\%$ when the reflux ratio increased further. The average effluent TN concentrations were approximately $3.08 \pm 0.14$ mg l$^{-1}$, $1.39 \pm 0.09$ mg l$^{-1}$ and $1.61 \pm 0.10$ mg l$^{-1}$, respectively. As a result, the optimum reflux ratio was found to be 2.

### 3.1.4. Performance of single-, two- and three-stage UODs

The UOD system was a multi-stage serial intermittent up-flow sludge reactor in which each stage integrated oxic/anoxic/anaerobic processes. The pollutant concentration decreased gradually from the first stage to the last stage, leading to a gradient distribution of DO concentration. The performance of a single-, two- and three-stage UODs was investigated for 60 days (electronic supplementary material, figure S4) with the optimum waterfall height (10 cm), up-flow rate (3.5 m h$^{-1}$) and reflux ratio (2). Figure 5 illustrates the average values of removal efficiency and effluent concentration.

In these three reactors, the average COD removal efficiency and effluent COD concentration were almost constant at $84.33 \pm 2.48\%$ and $20.67 \pm 2.85$ mg l$^{-1}$, respectively. The average effluent NH$_4^+$-N concentration was close to $0.40 \pm 0.08$ mg l$^{-1}$ in the single-stage reactor, decreased to $0.21 \pm 0.06$ mg l$^{-1}$ in the two-stage reactor and then dropped to zero in the three-stage reactor. The NH$_4^+$-N removal efficiency gradually increased. Additionally, the NO$_X^-$-N concentration showed a decreasing tendency in the single-, two- and three-stage reactors; for example, the average effluent NO$_3^-$-N concentration was $1.69 \pm 0.10$ mg l$^{-1}$, $1.48 \pm 0.09$ mg l$^{-1}$ and $1.37 \pm 0.09$ mg l$^{-1}$, respectively. The presence of NO$_2^-$-N dropped from $0.39 \pm 0.04$ mg l$^{-1}$ to 0. Similarly, the average TN concentration in the effluent decreased from $2.49 \pm 0.14$ to $1.39 \pm 0.09$ mg l$^{-1}$, and the TN removal efficiency increased from $88.56 \pm 0.97$ to

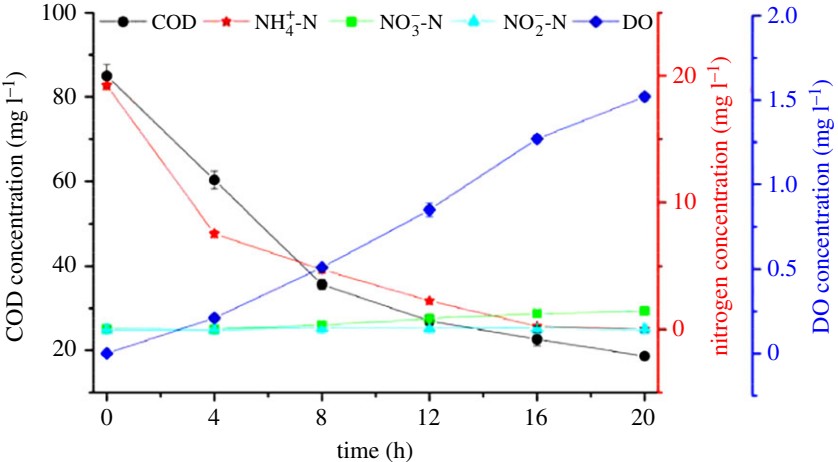

**Figure 6.** Main chemical compounds in the UOD during a typical cycle.

$93.63 \pm 0.60\%$. In figure 8, the structures of microbial community observed for three samples from the three-stage reactor were similar at the phylum and class levels. This result indicated that the enhancement in TN removal was possibly attributed to improvements in aeration and the alternative oxic/anoxic condition brought by the multi-stage reactor configuration. Hence, to improve the performance of UODs, multiple stages in series should be seriously considered.

## 3.2. N and COD conversion and DO concentration In a typical cycle

To explore the specific mechanism during the biological elimination of carbon and nitrogen, a typical 20 h cycle profile of the dynamics of the main chemical compounds was surveyed. A three-stage UOD reactor was employed under the optimal conditions. Figure 6 shows the corresponding concentrations of $NH_4^+$-N, $NO_2^-$-N, $NO_3^-$-N, COD and DO at 4 h intervals for 20 h after feeding in the first stage.

The influent concentrations of COD and $NH_4^+$-N were $140.33 \pm 1.53$ mg $l^{-1}$ and $21.22 \pm 1.01$ mg $l^{-1}$, respectively, and the concentration of $NO_3^-$-N in the return water was $1.41 \pm 0.02$ mg $l^{-1}$. During the feeding period, the DO concentration in the first stage was $0.00 \pm 0.01$ mg $l^{-1}$ and then rose to $0.21 \pm 0.02$ at 4 h after feeding. During these processes, the denitrifying bacteria used organic compounds to reduce nitrate, leading to the decline of COD and $NO_3^-$-N concentrations. The concentrations of COD and $NO_3^-$-N decreased to $60.33 \pm 2.08$ and $0.00 \pm 0.02$ mg $l^{-1}$, respectively. Meanwhile, the concentration of $NH_4^+$-N was dropped to $7.56 \pm 0.25$ mg $l^{-1}$ as ammonium was removed via normal nitrification catalysed by AOB, while the produced nitrite and the nitrate in return water were mainly denitrified by heterotrophic denitrifiers.

As the organic matter was slowly consumed, the DO concentration was gradually increased up to $0.85 \pm 0.04$ mg $l^{-1}$ at 12 h after feeding. During this process, the activity of NOB was greatly suppressed when the concentration of DO was less than $1.0$ mg $l^{-1}$. Thus, ammonium was mostly removed by partial denitrification and the ammonium concentration decreased to $2.26 \pm 0.10$ mg $l^{-1}$.

As the COD concentration further decreased to $18.67 \pm 0.58$ mg $l^{-1}$, the DO concentration was observed to be $1.52 \pm 0.02$ mg $l^{-1}$ at 20 h after feeding, leading to complete nitrification of ammonium to nitrate. The concentration of $NH_4^+$-N was decreased to $0.01 \pm 0.01$ mg $l^{-1}$, while the accumulation of $NO_3^-$-N reached up to $1.45 \pm 0.02$ mg $l^{-1}$ with a constant $NO_2^-$-N concentration of zero over the whole period.

## 3.3. Phosphorus removal efficiency of the UOD

To achieve greater degrees of phosphorous removal, the effects of the up-flow rate and iron dosing were investigated for 60 days (electronic supplementary material, figure S4). When $2$ g $l^{-1}$ iron scurf was settled in four UOD reactors with a rising up-flow rate (see §3.1.1), the average removal efficiency of TP increased from $85.45 \pm 2.46$ to $89.27 \pm 1.40\%$ and the average effluent TP concentration was decreased from $0.36 \pm 0.05$ to $0.27 \pm 0.02$ mg $l^{-1}$ in the effluent (figure 7a). This result could be attributed to the more effective oxidation of $Fe^{2+}$ to $Fe^{3+}$ with richer DO and a higher up-flow rate. Moreover, iron (1, 2 and 4 g $l^{-1}$) was added to the three UOD reactors (waterfall height of 10 cm). Figure 7b illustrates that under optimal operational conditions, iron dosing had a slightly higher effect

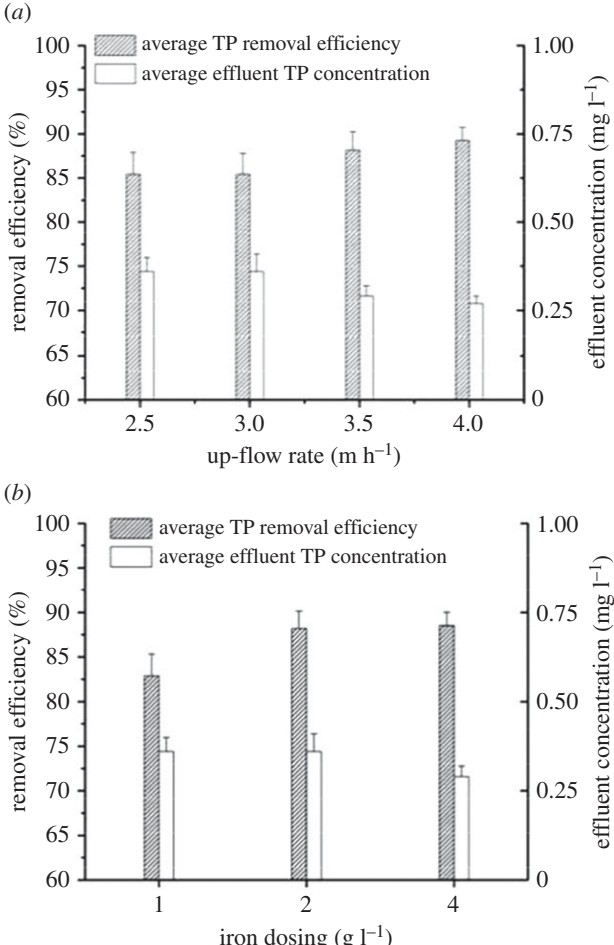

**Figure 7.** Phosphorous removal performances with the increasing up-flow rate (*a*) and iron dosing (*b*).

than the up-flow rate on phosphorous removal. Finally, the optimum conditions were determined to be an up-flow rate exceeding $3.5 \, \text{m h}^{-1}$ and an iron dosage greater than $2 \, \text{g l}^{-1}$.

It is hypothesized that iron–carbon micro-electrolysis could remove phosphorus through a variety of cooperative mechanisms, including microorganism activity, chemical precipitation and electrochemistry [38–42]. The proposed phosphorous removal scheme seems relatively simple and economical compared with conventional chemical methods.

These results revealed that the highly-efficient removal performances in UOD were impressive, since low-cost and saving energy treatment has drawn more attention. In comparative work, several post-treatments without external aeration or any other energy inputs were studied followed by UASB (as low cost and low energy) [24,43–45]. The data in table 2 show that the combined UASB/DHS (down flow hanging sponge) and DHNW (down flow hanging non-woven fabric) systems were more efficient in COD (greater than 90%) and BOD (greater than 90%) removal for municipal sewage treatment, while UMSR (up-flow microaerobic sludge reactor) exhibited a higher TN elimination (80.7%). The application of these configurations does not only produce methane to re-use, but also offers a sustainable solution for sewage treatment. However, the quality of effluent does not meet the strict discharge standards, especially in TN and TP removal. It was proven that UOD can supplement the higher requirement in the absence of methane production with excellent removal. Moreover, after UOD has been run over 2 years, we observed that the total amount of sludge was not changed that much. Therefore, we expected that the sludge age was long enough to cause the self-degradation in the reactor, resulting in the insignificant growth of sludge.

## 3.4. Microbial community structure

To further understand the biological mechanism in the UOD reactor, seed sludge (S0) and activated sludge (S1, S2 and S3 were collected in triplicate from the first, second and third stages of the UOD

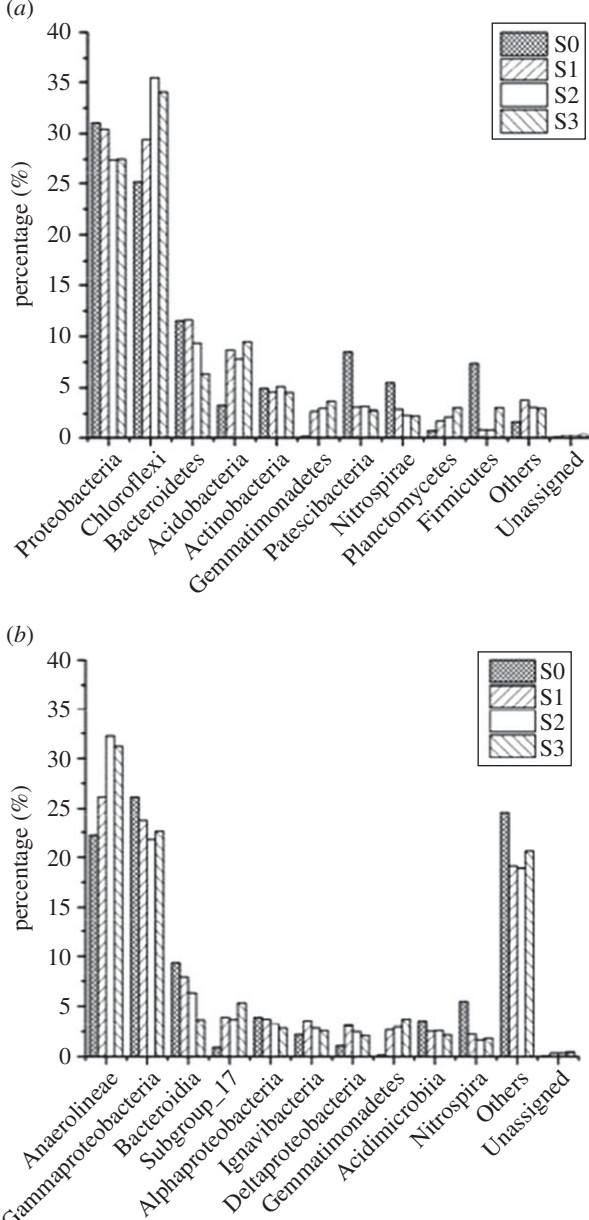

**Figure 8.** The bacterial community structures obtained at the phylum level (*a*) and class level (*b*).

reactor, respectively) were used for both DNA extraction and PCR amplification. The diversity indices of microbial communities observed in this study are shown in table 3. The corresponding coverage index (greater than or equal to 0.998) demonstrated that our sampling depth was sufficient enough. The numbers of sequencing reads, OTUs, Chao1 index, Simpson index and Shannon index in S1–3 were slightly higher than those in S0, indicating that the microaerobic environment could enhance the relative abundance and diversity of the bacterial community. The adaptation of the microbial community can effectively cometabolize environmental pollutants [8,20,30].

Figure 8 illustrates the identified bacterial structures and the relative abundances of the four samples at the phylum and class level. Figure 8*a* shows that the bacterial structure at the phylum level in S1–3 was different from that in S0. The dominant bacterial phyla within S0 appeared to be Proteobacteria (31.08%), Chloroflexi (25.14%), Bacteroidetes (11.52%), Acidobacteria (3.25%), Actinobacteria (4.97%), Patescibacteria (8.46%), Nitrospirae (5.48%) and Firmicutes (7.38%), while Proteobacteria (27.44–30.42%), Chloroflexi (29.39–35.43%), Bacteroidetes (6.35–11.58%), Acidobacteria (7.82–9.46%), Actinobacteria (4.54–5.13%), Gemmatimonadetes (2.69–3.67%), Patescibacteria (2.76–3.16%), Nitrospirae (2.28–2.89%) and Planctomycetes (1.78–3.02%) were present within S1–3.

**Table 2.** Comparison of efficiency of pollutant removal.

| parameters | UASB + DHS | | UASB + DHNW | | UMSR | | UOD | |
| | HRT 8 h | | HRT 6 h | | HRT 24 h | | HRT 20 h | |
| | influent | removal (%) | influent | removal (%) | influent | removal (%) | influent | removal (%) |
|---|---|---|---|---|---|---|---|---|
| COD (mg l$^{-1}$) | 532 ± 42.8 | 91.0 | 441 ± 21 | 90.7 | 70 ± 2.1 | 84.2 | 129 ± 17.2 | 84.3 |
| BOD (mg l$^{-1}$) | 240 ± 17.8 | 96.0 | 309 ± 17 | 93.2 | 70 ± 1.9 | — | 62 ± 7.5 | 90.3 |
| TN (mg l$^{-1}$) | — | — | 32 ± 7 | 62.5 | 70 ± 1.4 | 80.7 | 21 ± 1.3 | 93.6 |
| TP (mg l$^{-1}$) | — | — | — | — | — | — | 2.5 ± 0.3 | 89.2 |
| reference | Tandukar et al. [45] | | El-Khateeb et al. [43] | | Zhang et al. [24] | | | |

Seed sludge (S0) was collected in a limited-oxygen oxic tank during the daytime so it formed a symbiotic colony including aerobic, anaerobic and facultative bacteria. In response to long-term operation under microaerobic conditions, the proportions of Chloroflexi, Acidobacteria, Gemmatimonadetes and Planctomycetes in S1–3 increased dramatically in comparison to those in S0, but there was a decrease in the proportions of Proteobacteria, Bacteroidetes, Patescibacteria, Nitrospirae and Firmicutes. Moreover, Proteobacteria, Chloroflexi and Bacteroidetes were dominant phyla in all samples, which were in accordance with previous studies [8,10,12,13,16,17,30]. These phyla play important roles in carbon and nitrogen metabolism. Dedysh & Sinninghe-Damste [46] revealed the functional role of Acidobacteria, including the decomposition of biopolymers and participation in the cycling of carbon, iron and hydrogen. The relative abundance of Gemmatimonadetes was positively correlated with nitrite concentration, which was accumulated at low DO concentrations as reported in the literature [47,48]. NOB can be washed out under low-oxygen conditions, and they were broadly distributed among Nitrospira and Proteobacteria [6,12]. This distribution might be the reason why their proportions were reduced in S1–3. Other phyla, such as Actinobacteria and Patescibacteria, were related to the decomposition of organic substances and nitrogen cycling, respectively [49].

Differences between the four samples at the class level are illustrated in figure 8b. The predominant class within S1–3 was the anaerobic bacteria Anaerolineae (26.18–32.25%), which was higher than that in S0 (22.18%). Gammaproteobacteria, Bacteroidia, Subgroup_17, Alphaproteobacteria, Ignavibacteria, Deltaproteobacteria, Gemmatimonadetes, Acidimicrobiia and Nitrospira were observed as dominant classes, summing up to 74.75, 79.32, 79.62 and 77.89% for S0–3, respectively. A higher diversity of low-abundance bacteria (less than 1%) at the class level was observed in S1–3 than in S0. These data were consistent with the diversity indices.

The key functional groups at the genus level are shown in figure 9. During the long-term operation with low DO conditions, the total proportion of AOB was observed to rapidly increase from 0.48% in S0 to 1.82% in S1. Ellin6067 (a main species of AOB) had a lower ratio in S0 (0.37%) than S1 (1.49%). By contrast, the proportion of NOB was 5.47% in S0 and 2.24% in S1. Nitrospira, a common genus of NOB, had a higher proportion in S0 (5.47%) than S1 (2.23%). This result indicated that low DO was favourable for the enrichment of AOB and the bition of NOB in the UOD reactor, which were essential for achieving partial nitrification. The proportions of typical anaerobic ammonium-oxidizing bacteria (AnAOB), Candidatus_Brocadia, Candidatus_Jettenia and Candidatus_Kuenenia, changed from nearly zero in S0 to 0.14%, 0.06% and 0.36% in S1, respectively. The enrichment of these genera played a significant role in the elimination of the UOD reactor. Since nitrite acted as the electron acceptor and $CO_2$ served as the carbon source, AnAOB can convert ammonium into nitrogen gas [50]. Thus, anammox has been acknowledged as a more economical and effective process for treating wastewater. Heterotrophic denitrifying bacteria showed almost the same relative abundance (4.66–4.80%) in both samples, and 19 genera of denitrifying bacteria were identified, including Denitratisoma, Comamonas, Pseudomonas, Limnobacter, Ottowia, Aridibacter, Hyphomicrobium and Phaeodactylibacter. Ottowia had a higher abundance (1.23% in S0 and 1.00% in S1) than the other genera and is also regarded as a functionally important hydrolysis bacteria [51]. Hence, Ottowia not only participated in nitrogen removal but also degraded organic components [52]. Members of the

**Table 3.** The diversity of the microbial community. All experiments were performed three times to obtain the means with standard deviation.

| name | reads | OTUs[a] | ACE | Chao1 | Simpson | Shannon | coverage |
|------|-------|---------|-----|-------|---------|---------|----------|
| S0 | 50 433 ± 3263 | 892 ± 30 | 939.901 ± 31.469 | 959.401 ± 37.948 | 0.017 ± 0.004 | 5.089 ± 0.154 | 0.998 ± 0.000 |
| S1 | 51 154 ± 1621 | 983 ± 6 | 991.404 ± 5.702 | 995.841 ± 8.749 | 0.013 ± 0.001 | 5.540 ± 0.010 | 0.999 ± 0.000 |
| S2 | 51 277 ± 1789 | 980 ± 2 | 994.183 ± 6.619 | 999.757 ± 11.438 | 0.020 ± 0.002 | 5.347 ± 0.023 | 0.999 ± 0.000 |
| S3 | 52 325 ± 1616 | 973 ± 9 | 996.460 ± 20.229 | 1003.706 ± 18.920 | 0.031 ± 0.011 | 5.054 ± 0.249 | 0.999 ± 0.001 |

[a]OTUs are defined at a sequence similarity greater than or equal to 97%.

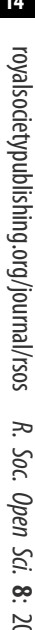

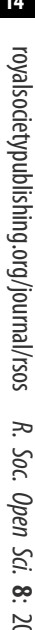

**Figure 9.** The specific bacterial community structure obtained at the genus level in S0 (*a*) and S1 (*b*).

dominant genera Comamonas, Pseudomonas, Limnobacter and Aridibacter are common in WWTPs. Comamonas is considered a complete denitrifier with Aridibacter under anoxic conditions, and these genera can use amino acids, polysaccharides and organic acids as carbon sources for denitrification

(a)

$$NH_4^+ \xrightarrow{\text{AOB}} NO_2^- \xrightarrow{\text{NOB}} NO_3^-$$

anammox → $N_2$

denitrification    denitrification → $N_2$

(b)

$$Fe \xrightarrow[-2e]{} Fe^{2+} \xrightarrow[-e]{O_2} Fe^{3+} \xrightarrow{PO_4^{3-}} FePO_4 \downarrow$$

**Figure 10.** Potential nitrogen (a) and phosphorous (b) conversion pathways in the UOD treatment process.

[53]. Additionally, Pseudomonas (0.36% in S0 and 0.22% in S1), Hyphomicrobium (0.40% in S0 and 0.33% in S1) and Phaeodactylibacter (1.06% in S0 and 0.89% in S1) are aerobic denitrifiers [54,55]. Notably, Denitratisoma was almost not detected in S0, but its relative abundance increased in S1 (0.58%). Denitratisoma can effectively convert nitrite into nitrogen gas under aerobic conditions [8]. Hydrogenophaga is an autotrophic denitrifier with hydrogen as an electron donor and its relative abundance increased to 0.13% in S1, which facilitated TN removal [40]. In addition, the total proportion of glycogen-accumulating organisms (GAOs) was higher in S1 (4.05%) than in S0 (1.94%). The competition for carbon sources between denitrifying bacteria and GAOs might be responsible for the significant enhancement in COD removal in the UOD reactor. By contrast, denitrifying polyphosphate-accumulating organisms (DPAOs) had a higher relative abundance in S0 (14.30%) and then the value dropped sharply to 0.25% in S1. Consistent results were obtained in that the concentrations of nitrate and nitrite were low in the UOD reactor and DPAOs and denitrifying bacteria competed for them as electron donors [13].

After long-term acclimation, the percentage of anaerobic fermentation bacteria increased from 0.27% (S0) to 1.24% (S1), which proved that the activated sludge granules and flocs can form a DO gradient under microaerobic conditions. Thus, various aerobes, anaerobes and facultative aerobes (such as AOB, NOB, AnAOB, autotrophic/heterotrophic denitrifying bacteria, GAOs, DPAOs, anaerobic fermentation bacteria, etc.) coexisted in a single UOD reactor and they cooperated to remove pollutants efficiently (figure 10).

# 4. Conclusion

To treat real municipal wastewater, a microaerobic UOD coupled with iron–carbon micro-electrolysis with waterfall aeration was designed. The simultaneous removal performance was considered at an HRT of 20 h and a constant temperature of $30 \pm 2°C$. The concentration of DO in the UOD could be slightly adjusted and the operational parameters were optimized, including the up-flow rate ($3.5$ m h$^{-1}$), waterfall height (10 cm), reflux ratio (2 : 1), number of stages (three) and iron dosing (a minimum amount of 2 g l$^{-1}$). The elimination efficiencies of COD, $NH_4^+$-N, TN and TP were obtained as $84.33 \pm 2.48\%$, $99.91 \pm 0.09\%$, $93.63 \pm 0.60\%$ and $89.27 \pm 1.40\%$, respectively, while the average effluent concentrations of COD, $NH_4^+$-N, TN and TP were $20.67 \pm 2.85$, $0.02 \pm 0.02$, $1.39 \pm 0.09$ and $0.27 \pm 0.02$ mg l$^{-1}$, respectively. Furthermore, the UOD reactor formed a DO gradient distribution within the whole process. The results obtained in high-throughput sequencing analysis demonstrated the coexistence of various functional groups of organisms in the microaerobic environment, which indicated that nitrogen was removed via multiple mechanisms, including nitrification, partial nitrification, denitrification and anammox. Thus, combining UOD with micro-electrolysis would be a useful platform for the elimination of nitrogen, COD and TP from municipal sewage and the potential nitrogen and phosphorous removal mechanisms are illustrated in figure 10.

Data accessibility. All data have been included in the manuscript and electronic supplementary material.

Authors' contributions. Z.-d.Z. designed and conducted the experiments, processed the data, discussed the results and wrote the paper. Q.-m.H. sampled and monitored COD. Q.L. and Y.-h.F. offered suggestions, materials and financial support. Y.Z. critically revised the manuscript. X.-h.W. engaged in the study design. All of the authors read and approved the submitted manuscript.

Competing interests. We declare no competing interest.

Funding. This work was financially supported by the Key Research and Development Project of Hainan Province (no. ZDYF2018107) and the Key Laboratory of Water Pollution Treatment and Resource Reuse of Hainan Province.

Acknowledgements. The authors are thankful to Haikou Wastewater Treatment Plant for assistance.

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
