## [Peer Review File · Royal Society Open Science]

Review History

RSOS-201887.R0 (Original submission)

Review form: Reviewer 1

Is the manuscript scientifically sound in its present form?

Yes

Are the interpretations and conclusions justified by the results?

Yes

Is the language acceptable?

Yes

Do you have any ethical concerns with this paper?

No

Have you any concerns about statistical analyses in this paper?

No

Recommendation?

Accept with minor revision (please list in comments)

Comments to the Author(s)

Zhao et al. designed a microaerobic up-flow oxidation ditch (UOD) coupled with micro-electrolysis for water purification. Development of cost-effective and environmentally sustainable methodologies for purification of municipal sewage water is an important research topic. The present work is impressive to warrant a publication in R. Soc. Open. Sci. journal.

Major issues

1. Authors should include the mechanism figure which describes the chemical reaction scheme leads to removal of nitrogen and phosphorous from municipal water.
2. Include the Comparison table showing the efficiency of present process over the reported methods in terms of cost effectiveness and green chemistry metrics.

Review form: Reviewer 2

Is the manuscript scientifically sound in its present form?

Yes

Are the interpretations and conclusions justified by the results?

Yes

Is the language acceptable?

Yes

Do you have any ethical concerns with this paper?

No

Have you any concerns about statistical analyses in this paper?

No

Recommendation?

Major revision is needed (please make suggestions in comments)

Comments to the Author(s)

A) The authors consider the TN as the following

$$TN = NH_3-N + NO_2-N + NO_3-N$$

What about organic nitrogen. The TN is calculated as the following:

$$TN = \text{Organic N} + NH_3-N + NO_2-N + NO_3-N$$

B) Could you compare your method with other methods such as:

- 1- Combining UASB and the "fourth generation" down-flow hanging sponge reactor for municipal wastewater treatment, Water Sci Technol, 2006;53(3):209-18. doi: 10.2166/wst.2006.095.
 - 2- The feasibility of using non-woven fabric as packing material for wastewater treatment. Desalination and Water Treatment. 2018;111:94-100.
 - 3- Integration of UASB and down flow hanging non-woven fabric (DHNW) reactors for the treatment of sewage water. Desalination and Water Treatment. 2019;164:48-55
- C) The resolution of the figures is very low. The font size is very small in some figures.

- D) Why did you neglect the BOD parameter in your work? This parameter measures the biodegradability of the organic loads in wastewater.
- E) What about the growth of sludge during the study?
- F) What is the number of samples analyzed during the study?

Decision letter (RSOS-201887.R0)

Dear Professor Lin:

Title: Pollutant Removal from Municipal Sewage by a Microaerobic Up-Flow Oxidation Ditch (UOD) Coupled with Micro-Electrolysis
Manuscript ID: RSOS-201887

The editor assigned to your manuscript has now received comments from reviewers. I apologise that this has taken longer than usual.

We would like you to revise your paper in accordance with the referee and Subject Editor suggestions which can be found below (not including confidential reports to the Editor). Please note this decision does not guarantee eventual acceptance.

Please submit your revised paper before 24-Mar-2021. Please note that the revision deadline will expire at 00.00am on this date. If we do not hear from you within this time then it will be assumed that the paper has been withdrawn. In exceptional circumstances, extensions may be possible if agreed with the Editorial Office in advance. We do not allow multiple rounds of revision so we urge you to make every effort to fully address all of the comments at this stage. If deemed necessary by the Editors, your manuscript will be sent back to one or more of the original reviewers for assessment. If the original reviewers are not available we may invite new reviewers.

On behalf of the Subject Editor Professor Anthony Stace and the Associate Editor Dr Nadia Martinez Villegas.

RSC Associate Editor:

Comments to the Author:

The research presented in this draft is original and of interest to RSOS audience, however some clarification regarding the mass balance of nitrogen is needed in addition to a comparison of your method with some others. Please read carefully each of the comments from the reviewers and address each of them.

RSC Subject Editor:

Comments to the Author:

(There are no comments.)

Reviewers' Comments to Author:

Reviewer: 1

Comments to the Author(s)

Zhao et al. designed a microaerobic up-flow oxidation ditch (UOD) coupled with micro-electrolysis for water purification. Development of cost-effective and environmentally sustainable methodologies for purification of municipal sewage water is an important research topic. The present work is impressive to warrant a publication in R. Soc. Open. Sci. journal.

Major issues

1. Authors should include the mechanism figure which describes the chemical reaction scheme leads to removal of nitrogen and phosphorous from municipal water.
2. Include the Comparison table showing the efficiency of present process over the reported methods in terms of cost effectiveness and green chemistry metrics.

Reviewer: 2

Comments to the Author(s)

A) The authors consider the TN as the following

$TN = NH_3-N + NO_2-N + NO_3-N$

What about organic nitrogen. The TN is calculated as the following:

$TN = Organic\ N + NH_3-N + NO_2-N + NO_3-N$

B) Could you compare your method with other methods such as:

1- Combining UASB and the "fourth generation" down-flow hanging sponge reactor for municipal wastewater treatment, Water Sci Technol, 2006;53(3):209-18. doi: 10.2166/wst.2006.095.

- 2- The feasibility of using non-woven fabric as packing material for wastewater treatment. Desalination and Water Treatment. 2018;111:94-100.
- 3- Integration of UASB and down flow hanging non-woven fabric (DHNW) reactors for the treatment of sewage water. Desalination and Water Treatment. 2019;164:48-55
- C) The resolution of the figures is very low. The font size is very small in some figures.
- D) Why did you neglect the BOD parameter in your work? This parameter measures the biodegradability of the organic loads in wastewater.
- E) What about the growth of sludge during the study?
- F) What is the number of samples analyzed during the study?

Author's Response to Decision Letter for (RSOS-201887.R0)

See Appendix A.

RSOS-201887.R1 (Revision)

Review form: Reviewer 2

Is the manuscript scientifically sound in its present form?

Yes

Are the interpretations and conclusions justified by the results?

Yes

Is the language acceptable?

Yes

Do you have any ethical concerns with this paper?

No

Have you any concerns about statistical analyses in this paper?

No

Recommendation?

Accept with minor revision (please list in comments)

Comments to the Author(s)

Figure 9 needs to be clear.

Decision letter (RSOS-201887.R1)

Dear Professor Lin:

Title: Pollutant Removal from Municipal Sewage by a Microaerobic Up-Flow Oxidation Ditch (UOD) Coupled with Micro-Electrolysis
Manuscript ID: RSOS-201887.R1

Thank you for submitting the above manuscript to Royal Society Open Science. On behalf of the Editors and the Royal Society of Chemistry, I am pleased to inform you that your manuscript will be accepted for publication in Royal Society Open Science subject to minor revision in accordance with the referee suggestions. Please find the reviewers' comments at the end of this email.

The reviewers and handling editors have recommended publication, but also suggest some minor revisions to your manuscript. Therefore, I invite you to respond to the comments and revise your manuscript.

Because the schedule for publication is very tight, it is a condition of publication that you submit the revised version of your manuscript before 16-Oct-2021. Please note that the revision deadline will expire at 00.00am on this date. If you do not think you will be able to meet this date please let me know immediately.

Supplementary files will be published alongside the paper on the journal website and posted on the online figshare repository (<https://figshare.com>). The heading and legend provided for each supplementary file during the submission process will be used to create the figshare page, so

please ensure these are accurate and informative so that your files can be found in searches. Files on figshare will be made available approximately one week before the accompanying article so that the supplementary material can be attributed a unique DOI.

Kind regards,
Dr Ellis Wilde
Publishing Editor, Journals

On behalf of the Subject Editor Professor Anthony Stace and the Associate Editor Dr Nadia Martinez Villegas.

RSC Associate Editor
Comments to the Author:
Thank you very much for the revised version of the manuscript. Changes were made up to the satisfaction of the reviewers. Only one minor change has to be made.

RSC Subject Editor
Comments to the Author:
(There are no comments.)

Reviewer comments to Author:
Reviewer: 2

Comments to the Author(s)
Figure 9 needs to be clear.

Author's Response to Decision Letter for (RSOS-201887.R1)

See Appendix B.

Decision letter (RSOS-201887.R2)

Dear Professor Lin:

Title: Pollutant Removal from Municipal Sewage by a Microaerobic Up-Flow Oxidation Ditch (UOD) Coupled with Micro-Electrolysis
Manuscript ID: RSOS-201887.R2

It is a pleasure to accept your manuscript in its current form for publication in Royal Society Open Science. The chemistry content of Royal Society Open Science is published in collaboration with the Royal Society of Chemistry.

Yours sincerely,
Dr Ellis Wilde
Publishing Editor, Journals

On behalf of the Subject Editor Professor Anthony Stace and the Associate Editor Dr Nadia Martinez Villegas.

RSC Associate Editor
Comments to the Author:
(There are no comments.)

Reviewer(s)' Comments to Author:

Appendix A

Response to Reviewers Comments (Manuscript ID: RSOS-201887)

Pollutant Removal from Municipal Sewage by a Microaerobic Up-Flow Oxidation Ditch (UOD) Coupled with Micro-Electrolysis

Zhen-dong Zhao^{1,2}, Qiang Lin^{1,*}, Yang Zhou³, Yu-hong Feng^{2,*}, Qi-mei Huang³ and Xiang-hui Wang¹

¹Key Laboratory of Water Pollution Treatment and Resource Reuse of Hainan Province, Key Laboratory of Natural Polymer Functional Material of Haikou City, College of Chemistry and Chemical Engineering, Hainan Normal University, Haikou 571158, China

²Analytical and Testing Center, Hainan University, Haikou 570228, China.

³School of Chemical Engineering and Technology, Hainan University, Haikou 570228, China

E-mail: linqianggroup@163.com; fengyuhong@hainanu.edu.cn

Dear editor and reviewers, we would like to thank the reviewers for their time and comments on our work for publication in this journal. They have provided a lot of thoughtful comments and suggestions. Our responses to their comments are provided below, and all changes in the revised manuscript are highlighted. In this file, the italic font indicates comments from reviewers, our responses are in blue font. The revised manuscript and figures with highlighted changes have been uploaded.

Reviewers' Comments to Author:

Reviewer: 1

Comments to the Author(s)

Zhao et al. designed a microaerobic up-flow oxidation ditch (UOD) coupled with micro-electrolysis for water purification. Development of cost-effective and environmentally sustainable methodologies for purification of municipal sewage water is an important research topic. The present work is impressive to warrant a publication in R. Soc. Open. Sci. journal.

Major issues

1. Authors should include the mechanism figure which describes the chemical reaction scheme leads to removal of nitrogen and phosphorous from municipal water.

According to the suggestion from the review, the possible mechanism was proposed in Figure 10. The revised parts have been highlighted by red color font in the revised manuscript (on Page 7).

2. Include the Comparison table showing the efficiency of present process over the reported methods in terms of cost effectiveness and green chemistry metrics.

According to this suggestion, we have compared the efficiency of pollutant removal in this work and that reported in the literature. The corresponding comparisons were summarized in Table 2. Please see our further discussions on Page 6. The revised parts have been highlighted by red color font in the revised manuscript.

Reviewer: 2

Comments to the Author(s)

A) The authors consider the TN as the following

$$TN = NH_3\text{- N} + NO_2\text{- N} + NO_3\text{- N}$$

What about organic nitrogen. The TN is calculated as the following:

$$TN = \text{Organic N} + NH_3\text{- N} + NO_2\text{- N} + NO_3\text{- N}$$

Thanks for the comment from reviewer. The influent concentration of TN were detected by a multi-parameter portable colorimeter according to APHA method on Page 3. The description have been highlighted by red color font in the manuscript.

In a biochemical reaction (anaerobic-aerobic alternation), organic nitrogen can be converted to ammonia nitrogen. In addition, organic nitrogen is also present in the solids, and organic nitrogen in the effluent is also related to suspended solids. Due to a precipitation step with low-speed circulation (0.5 m/h) in the MUOD reactor, the activated sludge forms a dense layer, and suspended solid is effectively intercepted. Moreover, the sludge swells slightly, and the drain has a specific T-shaped head. Under these conditions, the specific concentration of SS in the effluent is very low. Therefore, we presumed that the organic nitrogen content in the effluent is close to zero and can be ignored. In this case, the TN of effluent can be approximately estimated by the equation of $TN = NH_3\text{- N} + NO_2\text{- N} + NO_3\text{- N}$.

To further reply this question, we added the above discussion on Page 3. Please see the highlights by red color font in the revised manuscript.

B) Could you compare your method with other methods such as:

1- Combining UASB and the "fourth generation" down-flow hanging sponge reactor for municipal wastewater treatment, Water Sci Technol, 2006;53(3):209-18. doi:

10.2166/wst.2006.095.

2- *The feasibility of using non-woven fabric as packing material for wastewater treatment. Desalination and Water Treatment. 2018;111:94-100.*

3- *Integration of UASB and down flow hanging non-woven fabric (DHNW) reactors for the treatment of sewage water. Desalination and Water Treatment. 2019;164:48-55*

According to the review's suggestion, the comparisons of efficiency of pollutant removal with that reported in the literature were discussed (Page 6) and the comparison results were provided in Table 2. The revised parts have been highlighted by red color font in the revised manuscript.

C) The resolution of the figures is very low. The font size is very small in some figures.

Figures have been revised with higher resolution and larger font size.

D) Why did you neglect the BOD parameter in your work? This parameter measures the biodegradability of the organic loads in wastewater.

According to the suggestion, BOD has been added in Tables 1 and 2.

E) What about the growth of sludge during the study?

After the reactor was started, the continuous growth and accumulation of biomass would reach an equilibrium state. Especially, after the reactor has been run over 2 years, we observed that the total amount of sludge was not changed that much. During our test period, no excess sludge was discharged from the reactor. Therefore, we expected that the sludge age was long enough to occur the self-degradation in the reactor, resulting in the insignificant growth of sludge. In this case, the growth of sludge was presumed to be ignored in the reactor. So the growth of sludge was not tested.

To reply this question, we have added the above discussion on page 6. The revised parts have been highlighted by blue color font in the revised manuscript.

F) What is the number of samples analyzed during the study?

To optimize the performance, totally, 60 samples were collected for each experimental condition, and a total of 378 samples were collected. We have added the sample numbers on page 3. The revised parts have been highlighted by blue color font in the revised manuscript.

Appendix B

Response to Referees Comments (Manuscript ID: RSOS-201887.R1)

Pollutant Removal from Municipal Sewage by a Microaerobic Up-Flow Oxidation Ditch (UOD) Coupled with Micro-Electrolysis

Zhen-dong Zhao^{1,2}, Qiang Lin^{1,*}, Yang Zhou³, Yu-hong Feng^{2,*}, Qi-mei Huang³ and Xiang-hui Wang¹

¹Key Laboratory of Water Pollution Treatment and Resource Reuse of Hainan Province, Key Laboratory of Natural Polymer Functional Material of Haikou City, College of Chemistry and Chemical Engineering, Hainan Normal University, Haikou 571158, China

²Analytical and Testing Center, Hainan University, Haikou 570228, China.

³School of Chemical Engineering and Technology, Hainan University, Haikou 570228, China

E-mail: linqianggroup@163.com; fengyuhong@hainanu.edu.cn

Dear editors, referees and the Royal Society of Chemistry, we are grateful to you for the acceptance of manuscript for publication in Royal Society Open Science. Once again, thank you very much for your suggestions. Our responses to their comments are provided below, and all changes in the revised manuscript are highlighted. In this file, the italic font indicates comments from referees, our responses are in blue font. The revised manuscript and figures have been uploaded.

Comments to Author:

RSC Subject Editor

Comments to the Author:

(There are no comments.)

Reviewer comments to Author:

Reviewer: 2

Comments to the Author(s)

1. Figure 9 needs to be clear.

Figure 9 have been revised with higher resolution and larger font size.